# Graph Generation via Temporal-Aware Biased Walks

**Resul Tugay**                                                                  *resultugay@atauni.edu.tr*
*Department of Artificial Intelligence and Data Engineering*
*Ataturk University, Turkey*

**Eren Olug**                                                                        *olug20@itu.edu.tr*
*Department of Artificial Intelligence and Data Engineering*
*ITU AI Research and Application Center*
*Istanbul Technical University, Turkey*

**Elif Ak**                                                                            *elif.ak@mun.ca*
*Faculty of Engineering and Applied Science*
*Memorial University, Canada*

**Kiymet Kaya**                                                                     *kayak16@itu.edu.tr*
*Department of Computer Engineering*
*ITU AI Research and Application Center*
*Istanbul Technical University, Turkey*

**Sule Gunduz Oguducu**                                                          *sgunduz@itu.edu.tr*
*Department of Artificial Intelligence and Data Engineering*
*ITU AI Research and Application Center*
*Istanbul Technical University, Turkey*

**Reviewed on OpenReview:** *https://openreview.net/forum?id=lDnMlhk3aw*

## Abstract

Some real networks keep a fixed structure (e.g., roads, sensors and their connections) while node or edge signals evolve over time. Existing graph generators either model topology changes (i.e., edge additions/deletions) or focus only on static graph properties (such as degree distributions or motifs), without considering how temporal signals shape the generated structure. By approaching the problem from an unconventional perspective, we introduce TANGEM, that integrate a temporal similarity matrix into biased random walks, thereby coupling signals with structure to generate graphs that highlight patterns reflecting how nodes co-activate over time. We evaluate TANGEM using an approach that separates structural fidelity (clustering, spectral metrics) from downstream temporal consistency, allowing us to clearly isolate the impact of the topology generator itself. In structural benchmarks, TANGEM consistently outperforms strong baselines while remaining lightweight. These results show that adding attribute-guided bias to structural sampling produces more realistic graphs and establishes TANGEM as a basis for future models that further integrate evolving signals and structure.

## 1 Introduction

Graphs provide a powerful abstraction for representing complex relational data across diverse domains. These include social networks Leskovec & Mcauley (2012), biological and molecular structures Stark et al. (2005), recommender systems, and infrastructure networks such as roads, power grids, and computer systems

Li et al. (2018); Olug et al. (2024). Beyond representation, generative graph models, which learn underlying structural distributions, have become essential for drug discovery Liu et al. (2019), protein design, data augmentation Bas et al. (2024), and networked systems like the Internet of Things De et al. (2022).

In many of these settings, networks are not only structured but also temporal: nodes and edges may appear or disappear, and their signals (i.e., attributes) often fluctuate Rozemberczki et al. (2021). While several recent graph generation studies (e.g., TagGen Zhou et al. (2020), TIGGER Gupta et al. (2022), DAMNETS Clarkson et al. (2022)) tackle the case where the graph topology itself evolves, far less attention has been given to the practical critical regime where the network topology remains fixed, but the node or edge attributes change substantially over time. For example, the road infrastructure of a city is largely static, yet traffic intensities evolve dynamically; similarly, sensor networks retain their physical layout while measurements vary continuously. These networks are commonly referred to as *spatio-temporal*, as they model data structures where a fixed topology is associated with time-varying measurements on nodes or edges Li et al. (2018); Zhao et al. (2020).

We argue that even with a fixed topology, temporal dynamics should shape generated connectivity patterns (e.g., recurring motifs or emphasized spectral modes) because time dictates how nodes co-activate. However, existing literature typically diverges into three separate tracks: (1) topological evolution focusing on edge additions and deletions Zhou et al. (2020); Gupta et al. (2022); Clarkson et al. (2022); (2) static generation focusing only on structural patterns Bojchevski et al. (2018); Simonovsky & Komodakis (2018); You et al. (2018b); Shi et al. (2020); and (3) forecasting, which predicts attributes on a fixed graph but fails to let those dynamics inform the underlying graph generation process Islam et al. (2024); Yu et al. (2018); Li et al. (2018).

To address this gap, we propose TANGEM (Temporally Attributed Network GEneration Model), a name inspired by "tandem" to reflect two components working together. TANGEM bridges attribute and structure through a temporally-aware biased random walk. We calculate a pairwise similarity matrix $\rho$ from historical node attributes and inject it as a time-aware bias into a second-order walk, which is then modeled autoregressively using a decoder-only Transformer architecture. This design enables temporal co-activation to steer the emphasis of specific motifs and spectral modes during graph generation, an aspect currently overlooked in the literature. To the best of our knowledge, TANGEM is the first graph generator that explicitly integrates temporal signals into the walk sampling procedure.

Our main contributions and findings are threefold. First, we introduce a similarity matrix $\rho$ which is computed from historical attributes and injected as a time-aware bias into the structural walk sampling procedure. This approach explicitly couples temporal co-activation with structural sampling to generate attribute-aware walks, providing more informative input for the Transformer model. Second, we demonstrate through empirical analysis that this attribute-aware sampling leads to significant performance gains in graph structure generation. Finally, we provide a comprehensive set of ablation and downstream studies that characterize the behavior of the proposed walk mechanism and describe its strengths and weaknesses across various environments and evaluation metrics.

## 2 Related Works

**Deep graph generators.** Modern deep graph generators address the limitations of classical probabilistic models (e.g., Erdos & Renyi (1959); Holland et al. (1983)) by learning structural distributions from data. These models are broadly categorized into generative adversarial networks (GANs) Bojchevski et al. (2018), variational auto-encoders (VAEs) Simonovsky & Komodakis (2018), and autoregressive models You et al. (2018b). Other approaches leverage reinforcement learning You et al. (2018a) and flow-based learning Shi et al. (2020), primarily for molecular graph generation. Concurrently, diffusion and score-based generators have shown rapid progress. Score models tackle the isomorphism issue by operating on set-valued states or exchangeable representations Niu et al. (2020); Jo et al. (2022), while diffusion-style methods such as DiGress Vignac et al. (2023), discrete-state continuous time diffusion Xu et al. (2024), CoMeTh Xu et al. (2024), and DeFoG QIN et al. (2025) achieve high-fidelity graph generation, especially in molecular settings (i.e., big corpora with small graphs). However, these approaches focus almost exclusively on optimizing

static adjacency-level fidelity (or delta matrices Clarkson et al. (2022)) and typically ignore the temporal node/edge signals that co-evolve with graph structure.

**Temporal graph learning.** A parallel line of work focuses on temporal graph learning rather than generation, primarily predicting signals that evolve on fixed or slowly changing networks. Models such as T-GCN Zhao et al. (2020) and ST-GCN/DCRNN Yu et al. (2018); Li et al. (2018) combine graph convolutions with recurrent or diffusion mechanisms to forecast traffic and other time-series data. More recent efforts, such as DyGCL Islam et al. (2024), extend this to event prediction by learning dynamic representations with contrastive objectives. While these approaches demonstrate how temporal information, along with structure, improves forecasting, they are not designed for graph generation. Nevertheless, they highlight a bidirectional dependency between structure and attributes: (1) structure-to-signals, where topology informs forecasting Yu et al. (2018); Li et al. (2018), and (2) signals-to-structure, where temporal co-variation informs graph inference Dong et al. (2016); Thanou et al. (2016). Together, these studies underscore the necessity for models that jointly capture structure and temporal dynamics, serving as the direct motivation for our proposed approach.

**Dynamic graph generation with evolving topologies.** Another parallel line of work focuses on generating dynamic graphs by modeling topological evolution over time. Temporal interaction models, such as TagGen and TIGGER generate timestamped edges via temporal random walks and inductive embeddings, explicitly adding or deleting edges Zhou et al. (2020); Gupta et al. (2022). Similarly, DAMNETS predicts edge additions, removals, or persistence between snapshots using a learning delta matrix Clarkson et al. (2022). While motif-oriented approaches like DyMOND replicate evolving patterns, they still treat edge activity as the primary signal and generally ignore node or edge attributes Zeno et al. (2021). Collectively, these works prioritize structural changes and overlook node attributes as primary modeling targets.

While existing works often treat attributes as auxiliary features, TANGEM introduces a paradigm shift by jointly modeling the structure and attributes. Different from models such as TagGen or DAMNETS, which prioritize edge additions and deletions, TANGEM operates in the static topology with temporal node signals. Furthermore, TANGEM differs from walk-based adversarial methods like NetGAN Bojchevski et al. (2018). While NetGAN is designed to mimic the short, pure structural walks via a GAN, TANGEM utilizes a decoder-only Transformer to model longer sequences. This design allows to capture higher order structures that are lost when aggregating many short, pure structural walks. Consequently, TANGEM complements heavy, multi-graph generators like DiGress by offering a lightweight, attribute-aware architecture. The combination of feature-awareness and autoregressive modeling allows for generating large, semantically consistent graphs within scalable resources.

## 3  Methodology

In this section, we first define the task and then describe proposed method.

**Definition 1 (Temporally-Attributed Graph Generation)** *Given a temporally-attributed graph $\mathcal{G} = (\mathbf{G}, \mathbf{X})$, the primary task is to learn a generative mapping function $f : (\mathbf{G}, \mathbf{X}) \to \mathbf{G}'$, where $\mathbf{G}' = (V', E')$ is a generated static topology that preserves the joint structural-temporal characteristics of the input. While the core generation focuses on the graph structure, the framework maintains a node-correspondence mapping to allow the transfer of the attribute tensor $X$ onto the generated graph. This setup enables extended capabilities, such as Temporal Attribute Evolution, where a Spatio-Temporal Graph Convolutional Network (STGCN) can be employed to predict future states of $X_{T:}$ conditioned on the generated structure $\mathbf{G}'$.*

Prior research has demonstrated the effectiveness of biased random walks in capturing graph structure for both graph representation and generation tasks. Specifically, Bojchevski et al. (2018) proposed a GAN-based generative approach and highlighted that increasing the number and length of these walks improves the generative model's ability to learn. However, scale-free networks such as social and citation networks pose unique challenges as their number of edges grows faster than their nodes due to the preferential attachment. Consequently, the number of walks required for effective representation increases superlinearly with graph size, and even quadratically for certain scale-free networks. Moreover, sampling an excessive number of ordinary walks raises the risk of overfitting, where the model memorizes local patterns of the source graph

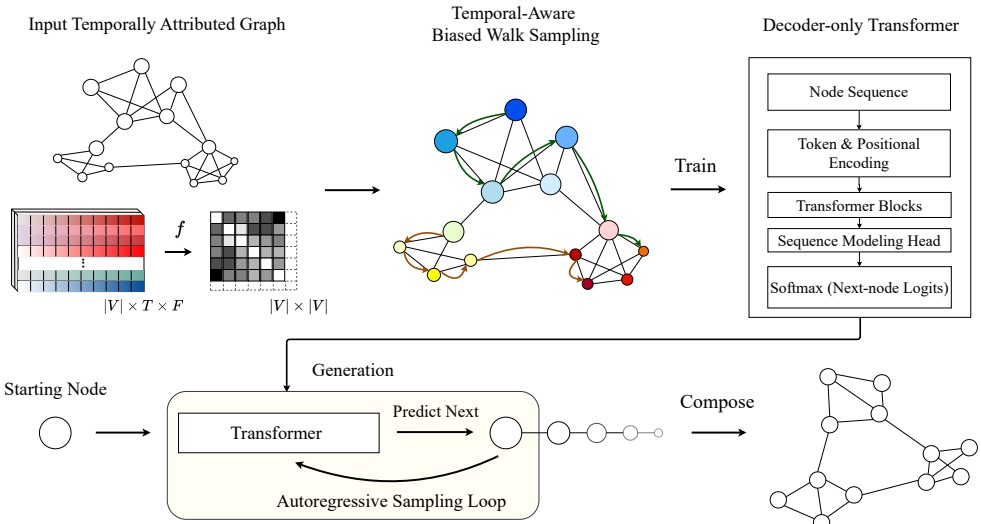

Figure 1: Overview of TANGEM. Attribute-aware biased random walks are sampled to capture rich connectivity and semantic structures (e.g., communities) within the graph. These walks train a Transformer model to generate node sequences. During inference, sequences are generated autoregressively and used to reconstruct the final graph structure.

rather than learning meaningful high-level patterns. These challenges underscore the need for strategically designed walks focused on generation density rather than pure quantity.

**Attribute-Aware Biased Random Walk**

The traditional second-order random walk proposed in Grover & Leskovec (2016) is guided by two parameters: the return parameter $p$ and the in-out parameter $q$. Suppose a walker has just traversed the edge from node $k$ to node $u$. The traditional unnormalized transition probability $\pi_{uv}$ from node $u$ to a neighbor $v$ is defined as:

$$\pi_{uv} = \alpha_{pq}(k, v) \tag{1}$$

where the structural search bias $\alpha$ is determined by the shortest path distance $\delta_{kv}$ between the previous node $k$ and the candidate node $v$:

$$\alpha_{pq}(k, v) = \begin{cases} \frac{1}{p} & \text{if } \delta_{kv} = 0 \\ 1 & \text{if } \delta_{kv} = 1 \\ \frac{1}{q} & \text{if } \delta_{kv} = 2 \end{cases} \tag{2}$$

The distance $\delta_{kv}$ is formulated as:

$$\delta_{kv} = \begin{cases} 0 & \text{if } k = v \\ 1 & \text{if } (k, v) \in E \\ 2 & \text{if } (k, v) \notin E \end{cases} \tag{3}$$

While this formulation effectively captures the topological skeleton of a network, these transitions are ultimately 'blind' to the rich, non-structural information embedded within node attributes. Tuning the $p$ and $q$ parameters alone is not sufficient for sampling the diverse, semantically-rich walks required for modern Transformer architectures. Low $p$ values cause the walker to remain constrained within a small local

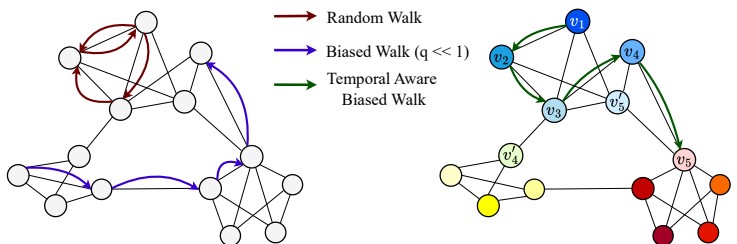

Figure 2: Comparison of random walk strategies. (Left) A uniform random walk moves stochastically, often resulting in redundant backtracking within a local neighborhood. (Middle) A biased exploratory walk ($q < 1$) prioritizes distant nodes, thereby reducing redundancy and encouraging broader structural discovery. (Right) Our attribute-aware biased walk balances exploration with semantic relevance, prioritizing nodes with high semantic similarity (indicated by similar colors) to capture both the global connectivity and local structures.

neighborhood, resulting in a lack of high-level global structures. While a low $q$ solves this by encouraging the DFS-like traversal, this exploration remains purely structural and ignores the underlying semantics within community structures. Consequently, there is a need for a strategy that prioritizes exploration while remaining semantically coherent.

To bridge this gap and obtain more meaningful sequences, we propose the integration of an attribute-bias $\rho$. We define an attribute similarity matrix $\mathbf{S} \in \mathbb{R}^{|V| \times |V|}$, where the entry $s_{uv}$ quantifies the semantic alignment of node trajectories. To maintain computational efficiency, $s_{uv}$ is computed only for existing edges:

$$s_{uv} = \begin{cases} f(\mathbf{x}_u, \mathbf{x}_v) & \text{if } (u, v) \in E \\ 0 & \text{otherwise} \end{cases} \tag{4}$$

where $f(\cdot, \cdot)$ is a similarity function acting on the feature vectors $\mathbf{x}_u$ and $\mathbf{x}_v$. The choice of $f$ is flexible, such that Dynamic Time Warping (DTW) similarity may be used for time-series attributes, while cosine similarity may be used for sparse feature vectors. We extend the node2vec transition weight by adding the attribute bias $\rho(u, v)$. The modified unnormalized transition probability $\pi'_{uv}$ is defined as:

$$\pi'_{uv} = \alpha_{pq}(k, v) \cdot \rho(u, v) \tag{5}$$

where the attribute bias is formulated as:

$$\rho(u, v) = \max(\lambda + (1 - \lambda)\, \hat{s}_{uv},\ \varepsilon) \tag{6}$$

Here, $\hat{s}_{uv} \in [0, 1]$ is the normalised similarity score obtained from $\mathbf{S}$, while $\lambda \in [0, 1]$ is a user-defined hyperparameter that balances the structural and feature-driven influence, and $\varepsilon > 0$ is a small value to prevent probabilities collapsing zero.

Finally, the transition probability is obtained by normalizing over the neighborhood $\mathcal{N}(u)$ of the current node $u$:

$$P(v \mid u) = \frac{\pi'_{uv}}{\sum_{z \in \mathcal{N}(u)} \pi'_{uz}} \tag{7}$$

For intuition, Fig. 2 illustrates how temporal features are incorporated into an explorative ($q < 1$) walk sampling procedure. Suppose the walker is at $v_3$; the exploratory bias reduces the chance of visiting $v_1$ or $v'_5$, but does not distinguish between $v_4$ and $v'_4$; here, the temporal bias favors $v_4$, which lies in a temporally coherent region. Conversely, when the walker is at $v_4$, the exploratory bias discourages revisiting $v'_5$, which

is already connected to the previous node $v_3$, and instead encourages moving forward to $v_5$, which promotes exploration.

**Autoregressive Generation via Sequence Modelling**

The autoregressive graph generation paradigm captures the complex joint probability of nodes and edges by leveraging recurrent or attention-based architectures You et al. (2018b); Bojchevski et al. (2018); Liao et al. (2019). Unlike one-shot generators, which often suffer from quadratic complexity relative to the number of nodes Tavakoli et al. (2017), autoregressive models scale more effectively to large graphs. We employ a Transformer architecture to learn the marginal distribution of nodes from feature-aware biased walks. This approach enables the model to look beyond immediate local neighborhoods and capture higher-order structure implicitly defined by node attributes and topology.

During inference, we generate graphs by sampling a sequence of nodes conditioned on previously generated tokens. The joint probability of generating a sequence $\mathbf{s}$ is composed as in Eq. 8, where $v_1$ is the initial node and $\mathbf{s}_{<m}$ represents the prefix generated so far.

$$P(\mathbf{s}) = P(v_1) \prod_{m=2}^{M} P(v_m \mid \mathbf{s}_{<m}) \tag{8}$$

The hyperparameter $M$ controls the sequence length. Unlike methods that rely on aggregating many short walks Bojchevski et al. (2018), we instead generate a small number of longer sequences ($k$). This design choice is to leverage the Transformer's ability to capture long-range dependencies within a single sequence. Because these $k$ sequences are generated independently, the final graph structure is formed by the union of all nodes and edges encountered across the $k$ walks. This also allows for generation graphs with more than one connected component.

Notably, the resulting sequences directly encode the connectivity: if a single sequence of length $M$ is used, the resulting graph contains at most $M$ nodes. The node count is exactly $M$ only if the resulting structure is a path graph. Otherwise, the repeated vertices within the sequence introduce higher-level structures such as cycles and clusters.

## 4 Experiments

We evaluate TANGEM across a diverse types of synthetic and real-world datasets, benchmarking its structure generation performance against competitive baselines using five key structural fidelity dimensions: degree distribution, clustering coefficients, spectral properties, motif counts, and orbit statistics. To investigate the model's specific mechanisms and robustness, our experiments address the following research questions:

- **RQ1 (Structural Generation Performance):** How does TANGEM compare to competitive generative baselines in capturing global and local structural properties (measured via MMD) across diverse datasets?

- **RQ2 (Ablation Study):** How do different walk sampling strategies influence the generative quality?

- **RQ3 (Sensitivity to Feature Distribution):** How does the feature-guidance mechanism adapt to varying feature distribution settings, ranging from highly homophilic to strictly heterophilic topologies?

- **RQ4 (Downstream Utility):** How does the synthesized topology $G'$ support effective temporal attribute evolution when utilized by a Spatio-Temporal Graph Convolutional Network (STGCN) Yu et al. (2018) for future state prediction ($X_{t+1}$)?

- **RQ5 (Scalability):** How does TANGEM scale in terms of training time and memory consumption relative to DiGress, a diffusion-based graph generator?

### 4.1 Experimental Setup

#### 4.1.1 Datasets

We evaluate TANGEM on both synthetic and real-world datasets with varying sizes and structural properties (e.g., community, grid-like, path-like).

**IBB network.** The IBB graph dataset Olug et al. (2024) consists of real-world road traffic data collected in Istanbul. The network is composed of small to mid-sized sub-networks, each corresponding to a specific district. The aggregated network contains $|V| = 2,451$ nodes and $|E| = 3,667$ edges, where nodes represent individual sensors and edges denote the road connections. To ensure the scalability of baseline models, we've utilized two sub-networks for the experiment in RQ1, while the aggregated network is utilized for other research questions. We utilized the *traffic density* as the primary temporal node attribute.

**PEMS-04 network.** The PEMS-04 Chen et al. (2001) traffic dataset was collected in a major metropolitan area in California. The network contains $|V| = 307$ nodes and $|E| = 338$ edges. For experiments, we utilize the largest connected component, which contains $|V| = 237$ nodes and $|E| = 280$ edges.

**Citeseer network.** While our primary focus is on spatiotemporal networks, we include the CiteSeer (CS) Sen et al. (2008) network to assess whether the proposed attribute-aware walk mechanism generalizes to static homophily-based relationships. In that network, nodes represent documents and edges represent citation relationships between them. Each node is characterized by a $3,703$-dimensional sparse binary feature vector representing a bag-of-words encoding of the document's content. Following NetGAN Bojchevski et al. (2018), we utilize the largest connected component, which contains $|V| = 2120$ nodes and $|E| = 3679$ edges.

#### 4.1.2 Baseline Models

We benchmarked TANGEM against both classical and modern graph generators. As traditional baselines, we used the Erdős–Rényi (E–R) model (Erdos & Renyi, 1959), which forms edges with a fixed probability, and the Barabási–Albert (B–A) model (Barabási et al., 2002), which reproduces scale-free networks via preferential attachment. For deep learning baselines, we included GraphVAE Simonovsky & Komodakis (2018), a variational autoencoder for latent-variable graph reconstruction; GraphRNN You et al. (2018b), an autoregressive model that sequentially generates nodes and edges using an RNN; GRAN Liao et al. (2019), a method that combines block-wise autoregression and self-attention; NetGAN Bojchevski et al. (2018), a method that uses random walks and adversarial training to learn a generative model; DiGress Vignac et al. (2023), a diffusion-based method that denoises the graph to produce high-quality samples. Since GraphRNN, GRAN, and DiGress require a dataset of multiple graphs for robust training, we employed a controlled data augmentation strategy via a low probability edge dropping ($\approx 3\%$). While doing that, we ensure that the core structure remains unaffected (e.g., connectivity).

#### 4.1.3 Evaluation Metrics

For structural evaluation, we follow the prior work You et al. (2018b); Liao et al. (2019); Vignac et al. (2023) and evaluate the generated graphs by calculating Maximum Mean Discrepancy (MMD) between generated and original graphs across key structural properties: degree distribution, clustering coefficient, spectral characteristics, motif count, and orbit count. For each property, we compute MMD using a Gaussian kernel to measure the divergence between distributions, ensuring a robust evaluation of both local and global features. Additionally, we utilize visual inspection.

### 4.2 Structural Generation Performance (RQ1)

Table 1 presents a comparison of various graph generative models, including two variants of our approach. TANGEM-Plain serves as a baseline ablation, where the Transformer is trained on uniform random walks rather than attribute-aware biased random walks. According to the results, TANGEM shows strong performance against baselines across all structural metrics. On the IBB1 and PEMS04 datasets, which are arguably sparse networks, TANGEM surpasses all methods in almost all metrics. This suggests that TANGEM is effective at modelling highly sparse graphs. However, some caution is needed since such paths could be

Table 1: Comparison of TANGEM to other generative models across different datasets and metrics (lower values are better). TANGEM-Plain is trained on uniform random walks, while TANGEM is trained on biased and temporal-aware walks with tuned hyperparameters ($p$ and $q$). Bold numbers correspond to the best performance, while underlined entries are the second-best performance.

| | IBB1 | | | | | IBB2 | | | | |
|---|---|---|---|---|---|---|---|---|---|---|
| | Degree | Clustering | Spectral | Orbit | Motif | Degree | Clustering | Spectral | Orbit | Motif |
| E-R Generator | 0.1464 | 0.0508 | 0.0862 | 0.1887 | 0.9817 | 0.1809 | 0.3898 | 0.0640 | 0.7971 | 1.2489 |
| B-A Generator | 0.4117 | 0.0001 | 0.2769 | 1.2684 | 1.5436 | 0.3033 | 1.4708 | 0.1262 | 1.2303 | 1.3013 |
| GraphVAE | 0.2952 | 1.2321 | 0.1592 | 1.3620 | 1.2369 | 0.3138 | 1.2748 | 0.1085 | 1.3574 | 1.2164 |
| NetGAN | 0.1316 | 0.0780 | 0.1037 | 0.1693 | 1.0169 | 0.2803 | 0.8705 | 0.0800 | 1.4898 | 1.7207 |
| GraphRNN | 0.0165 | 0.0527 | 0.0988 | 0.0057 | 0.2143 | 0.2243 | 0.1295 | 0.1266 | 0.3793 | 1.5659 |
| GRAN | 0.0153 | 0.0412 | 0.1180 | 0.0160 | 0.5055 | 0.0358 | 0.5573 | 0.0509 | 0.0259 | 0.5599 |
| DiGress | 0.0355 | 0.0071 | 0.1108 | 0.0073 | **0.0239** | 0.1417 | 0.0467 | 0.0655 | 0.0913 | 0.0987 |
| TANGEM-Plain | _0.0053_ | **0.0003** | _0.0720_ | _0.0032_ | 0.0530 | _0.0317_ | _0.0018_ | _0.0486_ | **0.0019** | **0.0329** |
| TANGEM | **0.0011** | _0.0002_ | **0.0425** | **0.0014** | _0.0251_ | **0.0101** | **0.0010** | **0.0378** | _0.0065_ | _0.0621_ |

| | Pems04 | | | | | CiteSeer | | | | |
|---|---|---|---|---|---|---|---|---|---|---|
| | Degree | Clustering | Spectral | Orbit | Motif | Degree | Clustering | Spectral | Orbit | Motif |
| E-R Generator | 0.1324 | 1.8028 | 0.1321 | 0.2152 | 1.2456 | 0.0805 | 1.9977 | 0.0705 | 1.9939 | 1.9534 |
| B-A Generator | 0.3191 | 1.8998 | 0.3617 | 1.1361 | 1.4508 | 0.0823 | 1.9881 | 0.0439 | 1.2782 | 1.3310 |
| GraphVAE | 0.2260 | 1.2625 | 0.1827 | 1.3945 | 1.2705 | - | - | OOM | - | - |
| NetGAN | 0.1213 | 1.3081 | 0.1656 | 0.0226 | 0.3007 | 0.0051 | **0.1371** | 0.0245 | 1.3514 | 1.1846 |
| GraphRNN | 0.0417 | 0.4823 | 0.1522 | _0.0036_ | _0.0259_ | - | - | OOM | - | - |
| GRAN | _0.0110_ | 0.7434 | _0.1191_ | 0.0133 | 0.1415 | 0.0131 | 1.7133 | 0.0245 | 1.3703 | 1.3763 |
| DiGress | 0.1383 | 1.0884 | 0.1570 | 0.4537 | 1.4593 | - | - | OOM | - | - |
| TANGEM-Plain | 0.0249 | _0.3355_ | 0.1220 | 0.0577 | 0.7944 | **0.0010** | _0.1520_ | _0.0083_ | _0.7460_ | _0.9144_ |
| TANGEM | **0.0005** | **0.0958** | **0.0368** | **0.0008** | **0.0164** | _0.0012_ | 0.1980 | **0.0075** | **0.4392** | **0.2621** |

memorized by the model due to the explorative nature of walks. To investigate this, we generate graphs with different sizes and assess their visual appearance. For IBB2, which is primarily a grid structure, no method clearly outperforms the others, though TANGEM remains competitive. On CiteSeer, NetGAN and GRAN are the comparable deep generative models, and TANGEM surpasses them on all metrics except clustering.

Following previous works, we further evaluate the results via visual inspection. Fig. 3 illustrates graphs generated by TANGEM alongside some baselines, highlighting TANGEM's ability to synthesize structures with diverse characteristics, such as path, grid, and community. Notably, TANGEM-generated CiteSeer (CS) graphs successfully capture community structures without memorizing training connections ($\approx 27\%$ novel edge ratio). In contrast, the NetGAN-generated graphs struggle to represent community structures, while the E-R baseline exhibits expected randomness.

### 4.3 Walk-Sampling Impact (RQ2)

Fig. 4 shows the effect of structural parameters, $p$ and $q$, on generation performance for IBB network. To isolate these structural effects, the feature-guidance mechanism was kept inactive. The results presented in Fig. 4 suggest that for the IBB network, DFS-like behavior with lower $q$ and higher $p$ values consistently results in lower MMD scores, implying better generation. We explain this as follows: DFS-like behavior encourages the walker to explore more distant connections, allowing the model to learn longer, more diverse structural paths. In contrast, models trained on BFS-like walks tend to generate sequences with high node repetitions. Consequently, even with a high token ($k$) and sequence count ($M$), the resulting graphs remain small and fail to accurately represent the structure of the source graph.

The Table 2 illustrates feature-similarity measures ($f$) including Pearson correlation, Cross-correlation, Cosine similarity, and Dynamic Time Warping (DTW) across two spatio-temporal traffic networks (PEMS04 and IBB) and one static attributed network (CiteSeer). For the Pems04 network, which is a small network, feature guidance does not contribute to the base case, which already receives low MMD scores. Conversely, the IBB spatio-temporal network, incorporating Cross-correlation as a measure of similarity, significantly outperforms the base case where no feature-guidance is applied. Furthermore, the Person correlation case

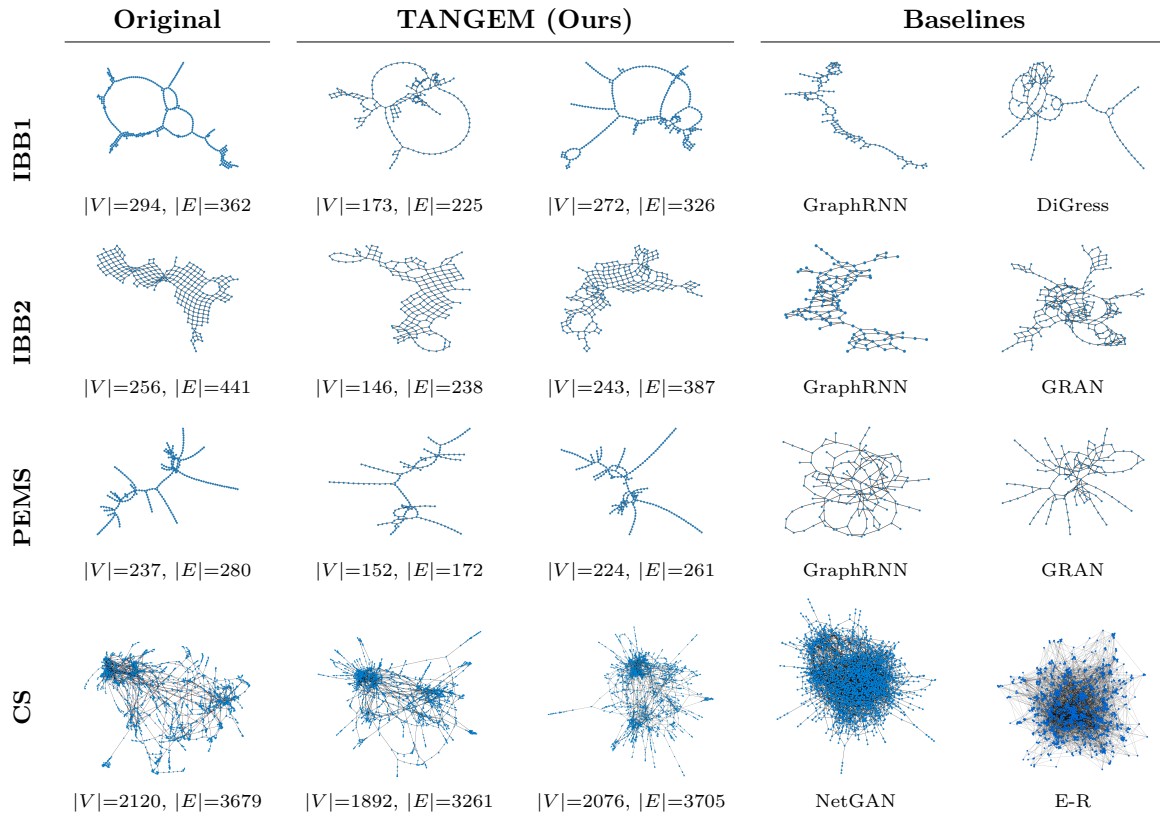

Figure 3: Qualitative comparison of graph generation results.

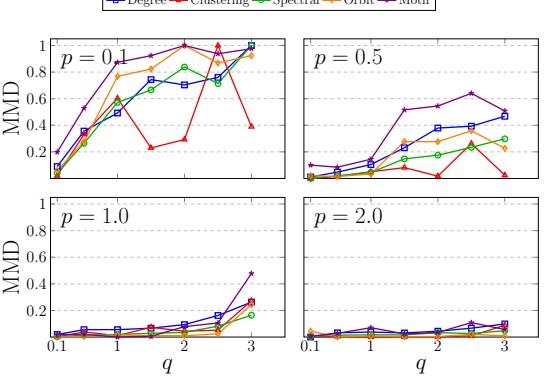

Figure 4: Analysis of structural bias parameters $p$ and $q$ (scores are normalized).

Table 2: Analysis of feature bias functions ($\times 10^{-2}$).

| | Method | Degree ($\downarrow$) | Orbit ($\downarrow$) | Motif ($\downarrow$) |
|---|---|---|---|---|
| **IBB** | Base | $1.18 \pm 0.39$ | $1.50 \pm 0.61$ | $33.62 \pm 14.77$ |
| | Pearson | $1.07 \pm 0.35$ | $1.42 \pm 0.47$ | $31.66 \pm 12.41$ |
| | Cross-Corr | $\mathbf{0.82 \pm 0.19}$ | $\mathbf{0.88 \pm 0.22}$ | $\mathbf{21.11 \pm 5.45}$ |
| | DTW | $1.27 \pm 0.17$ | $1.45 \pm 0.91$ | $\underline{29.64 \pm 13.20}$ |
| **Pems04** | Base | $0.02 \pm 0.01$ | $\mathbf{0.01 \pm 0.01}$ | $\mathbf{0.09 \pm 0.09}$ |
| | Pearson | $\mathbf{0.01 \pm 0.01}$ | $0.02 \pm 0.01$ | $0.39 \pm 0.26$ |
| | Cross-Corr | $0.02 \pm 0.02$ | $\underline{0.06 \pm 0.06}$ | $0.68 \pm 0.57$ |
| | DTW | $0.02 \pm 0.02$ | $0.04 \pm 0.03$ | $0.39 \pm 0.35$ |
| **CiteSeer** | Base | $\mathbf{0.06 \pm 0.01}$ | $67.34 \pm 24.64$ | $56.12 \pm 39.88$ |
| | Cosine | $\underline{0.07 \pm 0.01}$ | $\mathbf{46.23 \pm 23.72}$ | $\mathbf{14.01 \pm 10.99}$ |
| | Cross-Corr | $-$ | N/A | $-$ |
| | DTW | $-$ | N/A | $-$ |

also shows improvement over the base case, which suggests that even when the time-series nature of data is not fully leveraged, a simple linear correlation can enhance the generation performance. Finally, for the static CiteSeer network, applying Cosine similarity to the sparse bag-of-words feature set improves the overall graph generation quality.

Table 3: Mean purity (higher is better), coverage (higher is better), and percentage of noisy transitions (lower is better) across standard and proposed biased walks.

| | | (p,q) values | | | | | | | | |
| | | (0.5, 1.0) | | | (1.0, 1.0) | | | (1.0, 0.5) | | |
| Noise (%) | Method | Pur. | Cov. | N. T. | Pur. | Cov. | N. T. | Pur. | Cov. | N. T. |
|---|---|---|---|---|---|---|---|---|---|---|
| 0 | Standard | 0.737 | 0.582 | 0.000 | 0.718 | 0.660 | 0.000 | 0.699 | 0.724 | 0.000 |
| | Proposed | 0.738 | 0.580 | 0.000 | 0.719 | 0.658 | 0.000 | 0.700 | 0.721 | 0.000 |
| | *Change* | **+0.14%** | **-0.34%** | **0.00%** | **+0.14%** | **-0.30%** | **0.00%** | **+0.14%** | **-0.41%** | **0.00%** |
| 3 | Standard | 0.702 | 0.592 | 0.031 | 0.673 | 0.674 | 0.031 | 0.646 | 0.741 | 0.032 |
| | Proposed | 0.705 | 0.589 | 0.026 | 0.677 | 0.670 | 0.027 | 0.650 | 0.737 | 0.029 |
| | *Change* | **+0.43%** | **-0.51%** | **-16.13%** | **+0.59%** | **-0.59%** | **-12.90%** | **+0.62%** | **-0.54%** | **-9.37%** |
| 6 | Standard | 0.672 | 0.601 | 0.062 | 0.673 | 0.674 | 0.031 | 0.605 | 0.756 | 0.064 |
| | Proposed | 0.677 | 0.598 | 0.053 | 0.677 | 0.670 | 0.027 | 0.611 | 0.753 | 0.057 |
| | *Change* | **+0.74%** | **-0.50%** | **-14.52%** | **+0.59%** | **-0.59%** | **-12.90%** | **+0.99%** | **-0.40%** | **-10.94%** |
| 9 | Standard | 0.644 | 0.611 | 0.088 | 0.639 | 0.688 | 0.060 | 0.567 | 0.771 | 0.091 |
| | Proposed | 0.650 | 0.609 | 0.076 | 0.645 | 0.685 | 0.053 | 0.576 | 0.767 | 0.082 |
| | *Change* | **+0.93%** | **-0.33%** | **-13.64%** | **+0.94%** | **-0.44%** | **-11.67%** | **+1.59%** | **-0.52%** | **-9.89%** |
| 12 | Standard | 0.616 | 0.623 | 0.113 | 0.606 | 0.699 | 0.087 | 0.540 | 0.783 | 0.119 |
| | Proposed | 0.622 | 0.620 | 0.098 | 0.612 | 0.697 | 0.076 | 0.546 | 0.779 | 0.106 |
| | *Change* | **+0.97%** | **-0.48%** | **-13.27%** | **+0.99%** | **-0.29%** | **-12.64%** | **+1.11%** | **-0.51%** | **-10.92%** |

### 4.4 Sensitivity to Feature Distribution (RQ3)

#### 4.4.1 Semantic Filter Effect

While the homophily describes overall distribution of features in real-world networks (e.g., social and citation graphs), noisy, non-homophilic edges are often present. In this context, feature-guidance should prioritize transitions toward nodes with high feature alignment, effectively acting as a local adaptive filter that de-emphasizes noisy edges.

To validate this, we conducted an experimental analysis using the CiteSeer dataset, which is a strong homophilic network Zhu et al. (2020). We injected noisy edges into the original network at varying amounts (3%, 6%, 9%, and 12%). We then compared a standard structural random walk against our proposed feature-guided biased walk across the original and noisy networks. We evaluated the walks using three metrics: Label Purity (the percentage of nodes sharing the label of the starting node), Node Coverage (the ratio of unique nodes visited), and Noisy Transitions (N.T.) (the ratio of steps taken across noisy edges).

For this setup, we utilized the largest connected component of CiteSeer and used the cosine similarity on the 3,703-dimensional bag-of-words features. For each configuration, we generated 10 walks per node with a walk length of 10. The results are summarized in Table 3 with three $(p, q)$ hyperparameter settings including: BFS-like $(0.5, 1)$, uniform random $(1.0, 1.0)$, and DFS-like $(1.0, 0.5)$.

According to the results, across all noise configurations, shifting from a BFS-like setting to a DFS-like setting results in decreased Label Purity and increased Node Coverage. This aligns with the theoretical roles of $p$ and $q$. More importantly, results demonstrate that across all combinations, feature-guided biased walks consistently reduce the transition frequency of noisy edges. For the BFS-like case $((p, q) = (0.5, 1.0))$, noisy edge transitions drops by approximately $13\% - 17\%$, while this reduction ranges between $11\% - 12\%$ for the uniform case $((p, q) = (1.0, 1.0))$ and $9\% - 10\%$ for the DFS-like case $((p, q) = (1.0, 0.5))$. At the same time, the proposed walk mechanism increases purity across all combinations and noise levels, including the original network (0% noise). While we observe a consistent drop in coverage, it is mostly smaller than the gain in purity. Overall, these results validate the *semantic-filter effect* of the proposed walk mechanism in homophilic settings, where noisy connections are de-emphasized.

#### 4.4.2 Heterophilic Case

We examine the behavior of feature-guided walks within a heterophilic case. For this purpose, we construct a synthetic graph using a Stochastic Block Model (SBM) with four distinct communities. To analyze varying

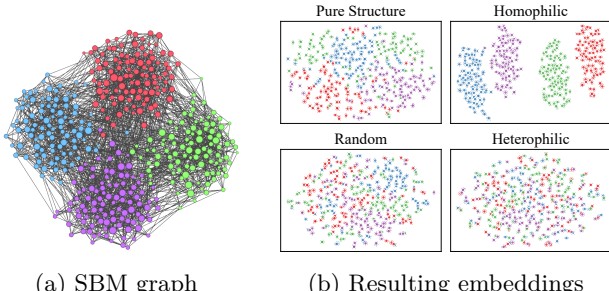

(a) SBM graph     (b) Resulting embeddings

Table 4: Scores across clustering metrics.

| Condition | NMI (↑) | ARI (↑) | Silh. (↑) |
|---|---|---|---|
| Pure Structure | 0.634 | 0.674 | 0.030 |
| Random | 0.606 | 0.656 | 0.027 |
| Homophily | **0.952** | **0.966** | **0.085** |
| Heterophily | 0.150 | 0.146 | 0.014 |

Figure 5: Analysis of structural and feature-based biases.

feature distributions, we generate homophilic, heterophilic, and random features over this structure. For each variant, node embeddings are generated using Node2Vec with walk parameters $(p, q) = (1.0, 0.1)$, biased toward transitions to nodes with lower Cosine distance. We evaluate the results using the K-Means clustering algorithm. Performance is measured against ground-truth communities using Normalized Mutual Information (NMI), Adjusted Rand Index (ARI), and Silhouette scores.

For the homophilic case, features are assigned based on a node's community membership. For heterophily, we apply graph coloring to each community separately; this ensures that connected nodes within the same community possess dissimilar features, while nodes in different communities may share similarities. Finally, we use a Gaussian distribution to assign random features to nodes, independent of any structural information.

The results presented in Table 4 indicate that in heterophilic settings, all evaluation metrics drop significantly compared to the topology-only (pure structure), homophilic, and random cases. This suggests that features misaligned with the graph structure force the walker to transition toward inter-community connections. Notably, the homophilic scenario yields the highest Silhouette, NMI, and ARI scores. These findings are visually confirmed by the t-SNE plots in Fig. 5b: while clusters are separated in the structural embeddings, they become increasingly blurred in the random case and almost indistinguishable in the heterophilic case. In contrast, the homophilic variant produces the most clear clusters.

While this experiment evaluates extreme homophilic/heterophilic cases, we believe that it provides insight into the impact of feature distribution on the walker's behavior. Consequently, while the current distance-based metrics (e.g., cosine, Euclidean, DTW) are optimized for homophilic structures, we have identified the integration of learnable distance metrics as a direction for future work to extend the method's applicability to heterophilic graphs.

## 4.5 Traffic Forecasting Downstream Utility (RQ4)

To evaluate the structural quality of the generated graph $\mathbf{G}'$, we conduct a downstream traffic forecasting experiment on the PEMS04 and IBB datasets. Firstly, we map the temporal attribute tensor $X$ onto the generated structure. While TANGEM inherently preserves node-correspondences between the source graph $\mathbf{G}$ and the generated structure $\mathbf{G}'$, for baseline methods, Erdős–Rényi, GraphRNN, NetGAN, and DiGress, we use a Hungarian-based structural matching algorithm to map the generated nodes to the original attribute indices.

In this setup, we use a Spatio-Temporal Graph Convolutional Network (STGCN) to predict future state $X_{t+1:t+12}$ conditioned on the generated graph $\mathbf{G}'$ and historical sequence $X_{t-12:t}$. By using $\mathbf{G}'$ as the underlying spatial topology, we test whether the generated structure preserves the underlying dependencies required for accurate temporal forecasting. We evaluate the predictive performance using three standard regression metrics: Mean Absolute Error (MAE), Root Mean Squared Error (RMSE), and Mean Absolute Percentage Error (MAPE). For clear benchmarking, we utilize several control cases: pure time-series prediction using an Autoregressive (AR) model that utilizes a simple neural network, STGCN with an identity

Table 5: Traffic forecasting results.

| Dataset | Metric | AR | | | STGCN | | | | |
|---|---|---|---|---|---|---|---|---|---|
| | | No Graph | Identity | E-R | GraphRNN | NetGAN | DiGress | TANGEM | Ground Truth |
| **PEMS04** | MAE | $28.2 \pm 0.1$ | $26.2 \pm 0.1$ | $\mathbf{24.4 \pm 0.1}$ | $\underline{25.4 \pm 0.1}$ | $25.6 \pm 0.1$ | $26.3 \pm 1.7$ | $\underline{25.4 \pm 0.1}$ | $25.4 \pm 0.2$ |
| | RMSE | $42.8 \pm 0.1$ | $39.8 \pm 0.1$ | $\mathbf{37.6 \pm 0.1}$ | $\underline{39.5 \pm 0.2}$ | $39.3 \pm 1.4$ | $39.8 \pm 1.9$ | $\underline{39.2 \pm 0.1}$ | $38.6 \pm 0.2$ |
| | MAPE | $19.8 \pm 0.2$ | $26.2 \pm 0.1$ | $\mathbf{14.4 \pm 0.1}$ | $16.3 \pm 1.0$ | $16.6 \pm 0.4$ | $16.0 \pm 1.1$ | $\underline{16.2 \pm 0.6}$ | $16.1 \pm 0.2$ |
| **IBB** | MAE | $26.9 \pm 0.1$ | $\underline{23.2 \pm 0.3}$ | $24.3 \pm 0.3$ | OOM | $23.6 \pm 0.4$ | OOM | $\mathbf{23.0 \pm 0.5}$ | $22.8 \pm 0.1$ |
| | RMSE | $48.4 \pm 0.0$ | $41.5 \pm 0.1$ | $41.5 \pm 0.3$ | OOM | $\underline{41.1 \pm 0.5}$ | OOM | $\mathbf{40.6 \pm 0.7}$ | $40.1 \pm 0.2$ |
| | MAPE | $57.0 \pm 0.1$ | $49.5 \pm 1.0$ | $52.6 \pm 2.5$ | OOM | $\underline{49.5 \pm 0.8}$ | OOM | $\mathbf{49.4 \pm 0.6}$ | $48.7 \pm 1.3$ |

adjacency matrix, and the ground truth scenario where the STGCN is trained on the original graph **G** and traffic features.

Table 5 presents the results. For the IBB network, TANGEM achieves the strongest performance following the ground truth. Conversely, the E-R case gets worse results than the STGCN with an identity adjacency matrix, suggesting that a corrupted structure results in poor predictive performance. Notably, GraphRNN and DiGress failed to scale to the IBB network ($|V| = 2451$, $|E| = 3667$).

For PEMS04, TANGEM still yields the closest scores to the ground truth case. Surprisingly, E-R graph outperforms the ground truth, which utilizes the actual graph structure. We hypothesize that, although the PEMS04 network is small, it is characterized by high diameter and extreme sparsity, which inherently restricts message passing between nodes. The E-R graph likely introduces 'shortcuts' that allow for more efficient information flow, thereby enhancing performance despite the structural alignment. Nevertheless, the structure generated by TANGEM shows the most similar behavior to the ground truth.

## 4.6 Scalability (RQ5)

We evaluate the computational cost of TANGEM against DiGress, which is a primary diffusion-based baseline model. While DiGress is effective for large datasets of small graphs (e.g., molecules), it faces significant scalability issues on larger graphs due to its $O(n^2)$ space complexity. Its iterative sampling process further scales linearly with the number of diffusion steps ($I$), resulting in an overall time complexity of $O(I \cdot n^2)$. For TANGEM, each token sampling step requires a linear cost of $O(n)$ [1]. Since $k$ sequences of $M$ tokens are independently parallelizable, the time complexity remains $O(M \cdot n)$, while the total computational work scales as $O(M \cdot \mathbf{k} \cdot n)$. Unlike diffusion methods, the space complexity remains $O(n)$ as no adjacency matrix is explicitly involved.

The total number of tokens generated, $M \cdot \mathbf{k}$, bounds the number of nodes in the resulting graph by $\min(M \cdot \mathbf{k}, n)$. In practise, due to the repeated node visits and cross-sequence overlap, the graph size is typically lower

---

[1]While standard Transformers exhibit $O(n^2)$ complexity, this can be linearized to $O(n)$ using kernel-based attention or low-rank approximations.

Table 6: Scalability Comparison of DiGress and TANGEM (OOM: Out of Memory, TO: Time Out)

| N | TANGEM | DiGress ($I = 100$) | DiGress ($I = 500$) |
|---|---|---|---|
| 50 | 0.10s / 18.0 MB | 0.44s / 41.7 MB | 2.20s / 41.7 MB |
| 100 | 0.21s / 19.0 MB | 0.48s / 86.7 MB | 2.39s / 86.7 MB |
| 200 | 0.43s / 19.0 MB | 0.88s / 269.4 MB | 4.43s / 269.4 MB |
| 400 | 0.86s / 25.0 MB | 3.30s / 972.8 MB | 16.48s / 972.8 MB |
| 800 | 1.72s / 28.0 MB | 12.42s / 3.8 GB | 62.12s / 3.8 GB |
| 1600 | 3.44s / 36.0 MB | 48.61s / 15.3 GB | TO |
| 3200 | 6.82s / 69.0 MB | OOM / TO | OOM / TO |

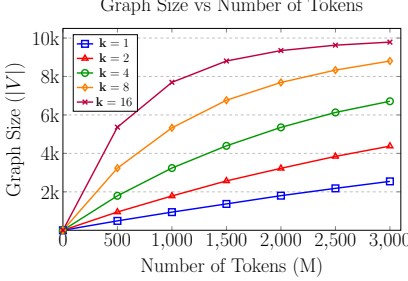

Figure 6: Generation parameters

than $M \cdot \mathbf{k}$. Fig. 6 illustrates the change in graph size as a function of $M$ and $\mathbf{k}$. These two hyperparameters provide flexible control over the generation size, which is particularly useful for large graph generation, where a single long sequence may be expensive and insufficient to capture the full structure.

Table 6 presents a comparative analysis of time and memory consumption for generating a single SBM graph across varying node counts with a fixed edge density ($|E| \approx 5 \times |V|$). The results indicate that while DiGress can generate graphs of several hundred nodes within reasonable limits, its requirements scale quadratically. For instance, generating a graph with 400 nodes requires 1 GB of memory. In contrast, TANGEM demonstrates superior scalability, maintaining linear growth in both time and space as the graph size increases. This experiment is conducted on an NVIDIA GeForce RTX 5090 GPU.

**Limitations.** While TANGEM is well-suited for domains such as transportation or sensor networks, the fixed structure assumption limits direct applicability to dynamic settings, such as social networks, where nodes and edges emerge frequently. Moreover, while the architecture is optimized for datasets consisting of a single or a few large-scale graphs, it remains unsuitable for datasets comprised of many small-sized graphs (e.g., molecular or protein datasets); in such cases, neither a union of all nodes nor matching nodes across graphs is feasible. Furthermore, TANGEM currently leverages a distance-based similarity signal to guide walk sampling. While this is effective for homophilic networks, it can be misleading in non-homophilic graphs where feature similarity does not imply semantic coherence. Future work could address this by utilizing learnable similarity functions where the similarity signal is derived from the learned representations of a method such as Graph Attention Network (GAT) Veličković et al. (2018). This would make it possible to determine more complex, meaningful connections in heterophilic environments.

## 5 Conclusion

We presented TANGEM, a generator for temporally attributed graphs with a fixed topology. By injecting the temporal similarity as a guidance to biased random walks, and then modeling these with a Transformer, TANGEM directly integrates signals and structure so that recurring co-activation patterns guide which motifs and spectral modes the generated graphs emphasize. Across benchmarks, it consistently improves structural metrics over strong static baselines. It has been shown that the attribute-aware biased walk mechanism, which is a key component of TANGEM, behaves as a noisy edge filter for homophilic graphs, while there is an open research area for applicability on heterophilic networks. We believe that TANGEM established a solid foundation for graph generation where structure and attributes are coupled.

## Reproducibility Statement

The main paper specifies the problem setting and evaluation separation (Sec. 3: the definition of the temporally-attributed graph and fixed-topology regime; Sec. 4: structural vs. downstream metrics) and defines all metrics (clustering, spectral MMD) and baselines. The Appendix details the topology generator (temporal-similarity construction, transformer walk model, and hyperparameters), ablations, and the routine feature-sampling setup used only for analysis. Our code repository [2] includes: implementation of TANGEM's topology module, exact training/evaluation scripts for each dataset, configuration files with all hyperparameters, random seeds, and run commands; data preprocessing pipelines for time-series benchmarks (including splits and normalization); and scripts to compute all reported metrics and figures. We also release preprocessed datasets, instructions for environment setup (including requirements and container specifications), and notes on hardware/runtime in the public repository.

### Acknowledgments

This research is supported by the Scientific and Technological Research Council of Turkey (TUBITAK) 1515 Frontier R&D Laboratories Support Program for BTS Advanced AI Hub: BTS Autonomous Networks and Data Innovation Lab project number 5239903 and TUBITAK 1501 project number 3250291.

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

## A Appendix

### A.1 Implementation Details of TANGEM

The TANGEM employs a transformer that processes sampled walks, treating graphs as sequences of tokens. The model uses learnable token and positional embeddings of size 64, with a context window defined by the block size. The architecture consists of 4 Transformer layers, each with 4 self-attention heads and a feed-forward network, connected through residual links and layer normalization. A dropout rate of 0.5 is applied. The final linear layer projects hidden states to the vocabulary space, and the model is trained with cross-entropy loss. During inference, the model generates graph sequences autoregressively by sampling from the predicted token distribution.

The AdamW optimizer is used for minibatch training, and each minibatch contains 128 tokenized walks. Depending on the input graph, walk length may differ (usually around 30-50). The default learning rate is

set as 0.001, and the maximum number of epochs is set as 10,000; early stopping is applied. All experiments except Scalability - RQ5 is conducted using Google Colab L4 accelerator for easy integration of baseline models. Scalability - RQ5 experiment is conducted on NVIDIA GeForce RTX 5090 GPU.

## A.2 Experimental Setup

We evaluated deep graph generation baselines using their official implementations and default settings: GraphRNN, GRAN, NetGAN, and DiGress. Since NetGAN relies on an older version of TensorFlow (1.x), we ran it locally, while the other models were executed in Google Colab notebooks for convenience. To provide training inputs for GraphRNN, GRAN, and DiGress, which require multiple graphs, we applied simple data augmentation (e.g., edge dropping) to generate additional graphs from the original one. For evaluation, each method generated 10 graphs, and we measured their similarity to the original graph using the MMD distance. The same evaluation procedure was applied consistently across all baselines.

## A.3 Notation Table

Table 7: Notation summary.

| Symbol | Description |
|---|---|
| $G = (V, E)$ | Underlying graph topology with node set $V$ and edge set $E$. |
| $X \in \mathbb{R}^{|V| \times T \times F}$ | Tensor of temporal node features; $T$ time steps, $F$ features per node per time step. |
| $x_i \in \mathbb{R}^{T \times F}$ | Temporal feature vector of node $i$. |
| $S \in \mathbb{R}^{|V| \times |V|}$ | Pairwise temporal similarity matrix computed from node features. |
| $S_{ij} = f(x_i, x_j)$ | Similarity between node $i$ and node $j$ defined by metric $f$. |
| $f(\cdot, \cdot)$ | Similarity metric (e.g., cosine similarity, correlation, RBF kernel). |
| $\rho(u, v) = S_{uv}$ | Temporal bias lookup function for nodes $u$ and $v$; extended in Eq. (7) to include second-hop neighbors. |
| $A$ | Adjacency matrix of the graph. |
| $L = D - A$ | Unnormalized Laplacian matrix; $D$ is diagonal degree matrix. |
| $p$ | Return parameter of Node2Vec-style second-order random walk. |
| $q$ | In–out (exploration) parameter of Node2Vec-style second-order random walk. |
| $\delta_{kv}$ | Shortest-path distance between nodes $k$ and $v$ (0, 1, or 2 for direct neighbors). |
| $\alpha_{pq}(k, v)$ | Search bias (Node2Vec) for moving from node $u$ to $v$ after visiting $k$. |
| $\alpha_q(k, v, u)$ | Modified search bias with temporal bias $\rho(u, v)$ (Eq. 5). |
| $s = \{v_1, \ldots, v_M\}$ | A node sequence (walk) of length $M$. |
| **k** | Number of sequences to sample during generation |
| $P(s)$ | Joint probability of generating sequence $s$ (Eq. 8). |
| $M$ | Sequence length / maximum number of tokens to generate. |
| $N(v)$ | Set of neighbors of node $v$. |
| $\lambda$ | Hyperparameter controlling influence of second-hop neighbors in $\rho(u, v)$. |

## A.4 Similarity Matrix

We utilize a pairwise similarity matrix $S \in \mathbb{R}^{|V| \times |V|}$ to integrate temporal node features, capturing their relationships and integrating this information into the walk sampling process as shown in Fig. 7. This allows the walks to reflect the more localized behavior of the nodes while maintaining exploration across the graph.

## A.5 More Visual Examples by TANGEM

We provide additional visual examples generated by TANGEM, showcasing its ability to capture complex patterns and temporal dynamics in graph structures, as seen in Table 8

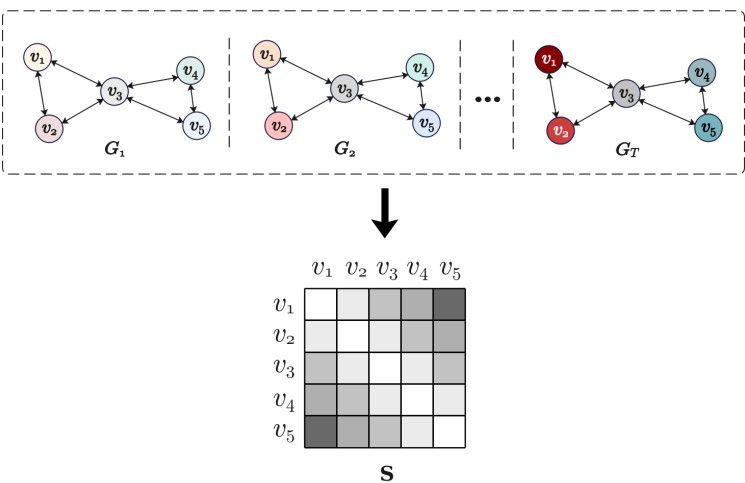

Figure 7: Similarity Matrix (S): similarity matrix using temporal node features across graph snapshots. Each entry reflects the similarity between nodes based on their temporal feature evolution over time.

Table 8: Original graph names are shown on the left. Each row has the original graph and four generated graphs by TANGEM.

| Original | Generated | | | |
|---|---|---|---|---|
| IBB1 | | | | |
| COM | | | | |
| PEMS | | | | |
| IBB2 | | | | |
| CiteSeer | | | | |

