# OpenReview forum: "Graph Generation via Temporal-Aware Biased Walks"
_TMLR — Accepted by TMLR_

### Review · Reviewer_3oxB · 2026-01-25

**Summary Of Contributions:**

In this paper, the authors propose TANDEM (Temporally Attributed Network GEneration Model), a neural network method that generates graphs based on random walk embeddings that are temporally biased. This bias is implemented following a Node2Vec-like procedure, where the walker is forced to explore nodes with high temporal affinity, i.e., nodes that share similar temporal signals. The walk is then used in a decoder only transformer for sequential generation, in the spirit of language modeling. Experiments over four different datasets are used to showcase the method's effectiveness, following the usual practice of graph generative modeling which is to compare distributions of the generated graphs's structural descriptors with the ground truth via MMD. The authors also provide experiments showing the effects of different random walk strategies in performance.

**Audience:**

Yes

**Audience Explanation:**

The work done in this paper is relevant to the broader GNN community, in particular to researchers working in graph generation. This community is regularly represented in TMLR, which means that the overall topic treated in the paper is relevant and in the scope of the journal.

**Broader Impact Concerns:**

The work does not tackle problems that require a Broader Impact Statement.

**Claims And Evidence:**

Yes

**Claims Explanation:**

There are different inconsistencies in the paper that the authors should consider addressing, as these greatly limit the contributions of the work. I will start by listing some positive aspects of the paper:

- The idea of biasing a random walk on a graph based on the signal for generation and not just structure is interesting.
- Showing that the above could be combined with a next token prediction objective is a good result, especially considering that autoregressive graph generation is not being considered a viable approach against modern graph diffusion models [1, 2]

As for the weaknesses:

- The definition of a temporally attributed graph is somewhat superfluous, considering the existence of spatio-temporal graphs [3]. In the context of the paper, the authors simply have a spatio-temporal structure where the spatial part is constant over time. Therefore, there is no need to provide a new definition for an existing structure. As an example, 3D pose skeleton estimation [4] is a computer vision task where graph connectivity remains the same (humans have the same skeleton) but the temporal attributes (position) change, which is exactly what Def. 1 considers to be a temporally attributed graph.

- The proposed random walk bias is somewhat contradictory given the listed assumptions. The authors first state that a node's signal trajectory is likely to mirror that of its neighbors, i.e., connected nodes share similar temporal evolution. Then, they add a temporal bias by basically moving to nodes whose temporal features are similar to the current node. But, starting from the previous assumption, this would mean moving to a node within the neighborhood, so effectively it is similar to using only the structure with a well tuned exploration parameter. Of course, in practice the behavior of the two walks would not be identical, but this would simply depend on the how homophilic the input graph is, which is something the authors also mention. To see this fact in practice, we can look at Fig. 4 / Tab. 4 (Appendix), where using BRW or BRW+ (temporally aware biased random walk) behave quite similarly on most structural properties besides for CiteSeer, but there doesn't seem to be a consistent advantage in using one over the other.

- The similarity function S is never characterized. This is quite the crucial aspect, because considering a simple variant such as say cosine similarity here completely breaks down the proposed topic of the paper. If temporal similarity is the key aspect, a similarity function that does not take it into consideration is just a way to bias the walk over *any* graph signal. To give a more practical summary, it is having TxD features or just D features is irrelevant if comparing two vectors over a vector space. Therefore, the authors seem to have no particular reason to consider just temporally attributed graphs. This point leads to another, which is that the authors always generate graphs that have fixed structure, therefore the idea of adding temporal bias is in fact simply adding a bias over the signal. Equivalently, it is as if the similarity matrix provides another adjacency matrix that connects the nodes based on feature similarity, so we can consider two graphs. This is something that could be applied to most graphs, not just ones with temporal attributes.

- The experimental section is too narrow to be able to draw conclusions. The authors use only four datasets that contain very small graphs besides CiteSeer, which in itself is not particularly large. This limits the ability to draw conclusions on scalability. Furthermore, a comparison with more sophisticated graph diffusion models such as [1,2] both in terms of MMD and scalability would offer great insights on the applicability of the model. As mentioned in my previous point, the proposed method is in no way "constrained" to spatio-temporal graphs, therefore the use of only 4 datasets is not sufficient for a paper that proposes a new generative model.

Therefore, most of the claims made in paper are not supported by clear and convincing evidence.

[1] Chen, Xiaohui, et al. "Efficient and Degree-Guided Graph Generation via Discrete Diffusion Modeling." Proceedings of Machine Learning Research (2023).

[2] Xu, Zhe, et al. "Discrete-state continuous-time diffusion for graph generation." Advances in Neural Information Processing Systems 37 (2024): 79704-79740.

[3] Du, Yuanqi, et al. "Disentangled spatiotemporal graph generative models." Proceedings of the AAAI Conference on Artificial Intelligence. Vol. 36. No. 6. 2022.

[4] Sampieri, Alessio, et al. "Pose forecasting in industrial human-robot collaboration." European Conference on Computer Vision. Cham: Springer Nature Switzerland, 2022.

**Requested Changes:**

The points listed here are reiterations of the weaknesses listed above. I urge the authors to consider all the aforementioned aspects carefully, but the list below contains a synthetic description of changes that would improve the paper:

- There are problems in the task definition: The authors mention they are doing "Temporally Attributed Network GEneration Model", model they are in fact just generating fixed topologies. At the current state, the method has no temporally-specific component, therefore the authors should consider reframing the problem under consideration and their contributions.

- The experimental evaluation is quite restricted. Consider adding datasets from [1,2,3] above and also comparing with modern diffusion-based methods [1,2]

- Provide a concrete characterization of S and how different choices could lead to potentially different results.

- Consider expanding the temporal bias of the walk beyond a very homophilic setting.

---

> ### Author Response · Authors · 2026-02-12
> **Additional Experiments on Walk Parameters and Graph Homophily**
>
> Thank you very much for the thoughtful and knowledgeable review. We take your points seriously and are willing to clarify and improve our work based on your suggestions. We would like to answer to some of your concerns, while working on others.
>
> #### Weaknesses
> **W1**: We initially avoided the term “spatio-temporal graph” because the definition provided by [1] describes a structure where “nodes and edges are embedded and evolve in a geometric space.” In contrast, our framework aligns with the definitions in [2,3], which describe a spatio-temporal graph as a **static topology with time-dependent features** rather than a dynamically evolving 3D structure.
>
> That being said, if the reviewer thinks “spatio-temporal graph” is the more standard or intuitive term for this context, we are open to incorporating it. We would value your further thoughts on that.
>
>
> **W2**: We thank the reviewer for their insightful observation regarding the potential overlap between our homophily assumption and the feature-guided (may be temporal or not) bias. We want to clarify why this bias is not redundant to the structural exploration, but rather serves as a filter for the neighborhood. While homophily is a global trend, our BRW+ bias filters local noise (e.g., cross-disciplinary citations in CiteSeer) that static structural parameters ($p, q$) cannot capture.
>
> New results in **RQ2** show improved cluster capturing in homophilic (CiteSeer is strongly homophilic [4]) networks.
>
> **W3**: We conducted new experiments (revised paper - RQ1)comparing four settings: (1) no temporal bias, (2) Pearson correlation, (3) cross-correlation, and (4) Dynamic Time Warping (DTW).
> Our results on Pems04 and IBB show that DTW consistently outperforms other methods. This is significant because DTW specifically accounts for temporal alignment and shifting, proving that the model relies on the temporal dynamics of the signal rather than just generic vector similarity. Furthermore, CiteSeer (non-temporal features) performs the best when Cosine similarity is used. We thank the reviewer for pointing this out.
>
> Additionally, our evaluation of the structural bias parameters ($p$ and $q$) shows that:
> - Increasing $p$ consistently improves the MMD score (generation quality).
> - Lowering $q$ (promoting exploration) further enhances performance.
>
> These two experiments together confirm that the model benefits most from explorative behavior and a temporally-aware similarity filter.
>
> **W4**: We agree that broader experimentation strengthens the paper’s conclusions. We have expanded our evaluation as follows:
>
> - Ablation of Walk Parameters (RQ1): We assessed the impact of $p$ and $q$ alongside various similarity functions (Pearson, DTW, etc.).
>
> - Homophilic vs. Heterophilic Analysis (RQ2, RQ3): We tested the method on both graph types. While feature-guided walks significantly improve cluster capturing in homophilic graphs, we candidly observed that the current iteration is less suited for heterophilic graphs.
>
>
> #### Requests
>
> **R1**:  We consider to revise the manuscript to clarify that while our primary focus is temporal data, the method is a general framework for **Attributed Graph Generation** where the similarity metric $S$ is the "plugin" that adapts the model to the attribute type (temporal or otherwise).
>
> **R2**: We shared the lastly added experiments in 'Response to Weakness 4'. Additionally, we are now working on:
> - Addition of two baseline models: BiGG [5] (autoregressive large graph generator), and Pard [6] (recent diffusion-autoregressive method).
> - We are also working on adopting new datasets from [7].
>
>
> **R3**: Please refer to Response to Weakness 2.
>
> **R4:** We have included an additional experiment involving an extreme-heterophily case (see **RQ3** in the revised manuscript). The results indicate that the current feature-guided biased random walk is optimized for homophilic structures. While this iteration focuses on leveraging graph homophily, adapting the framework for heterophilic graphs represents a promising direction for future research.

---

> > ### Author Response · Authors · 2026-02-12
> > **References:**
> >
> > [1] Du, Yuanqi, et al. "Disentangled spatiotemporal graph generative models." Proceedings of the AAAI Conference on Artificial Intelligence. Vol. 36. No. 6. 2022.
> >
> > [2] Li, Y., Yu, R., Shahabi, C., & Liu, Y. (2018). Diffusion convolutional recurrent neural network: Data-driven traffic forecasting. International Conference on Learning Representations (ICLR).
> >
> > [3] Zhao, L., Song, Y., Zhang, C., Liu, Y., Wang, P., Lin, T., Deng, M., & Li, H. (2019). T-GCN: A temporal graph convolutional network for traffic prediction. IEEE Transactions on Intelligent Transportation Systems, 21(9), 3848–3858.
> >
> >
> > [4] Zhu, J., Yan, Y., Zhao, L., Heimann, M., Akoglu, L., & Koutra, D. (2020). Beyond homophily in graph neural networks: Current limitations and effective designs. Advances in Neural Information Processing Systems, 33, 22619–22630.
> >
> >
> > [5] Dai, H., Nazi, A., Li, Y., Dai, B., & Schuurmans, D. (2020). Scalable deep generative modeling for sparse graphs. arXiv. https://arxiv.org/abs/2006.15502
> >
> > [6] Zhao, L., Ding, X., & Akoglu, L. (2024). Pard: Permutation-invariant autoregressive diffusion for graph generation. arXiv. https://arxiv.org/abs/2402.03687
> >
> > [7] Chen, Xiaohui, et al. "Efficient and Degree-Guided Graph Generation via Discrete Diffusion Modeling." Proceedings of Machine Learning Research (2023).

---

> > > ### Comment · Reviewer_3oxB · 2026-03-08
> > > **rebuttal response**
> > >
> > > I would like to thank the authors for their detailed response and the revised manuscript. Several new results have been added and at this point I believe that all my questions have been answered. I believe that the method's relevance is somewhat ambiguous, I believe that this version of the manuscript presents a sufficiently well-explored research questions regarding the generation of graphs with a time-varying signal. The authors also seem to have clearly delineated the contributions and limitations of the method and how it can be extended for future work. I have updated my review and would thus vote for an acceptance of the manuscript at the current stage.

---

### Review · Reviewer_FUvA · 2026-01-31

**Summary Of Contributions:**

The paper introduces TANGEM (Temporally Attributed Network GEneration Model), a method to learn a generator of graphs whose node attributes change with time, while their topology remains fixed. More specifically they learn a similarity matrix using historical signals  and inject it as a time-based bias into an autoregressive, transformer-based random walk, which leads to empirical gains on temporally
attributed benchmarks in structural fidelity.

**Audience:**

Yes

**Audience Explanation:**

The paper presents a new method for graph generation specially relevant for graphs whose node attributes change in time, while their topology remains the same, and then shows that such method is quite strong on 4 datasets commonly used in the literature, I believe there are clear applications of this technique that would be of interest to the audience of TMLR.

**Claims And Evidence:**

Yes

**Claims Explanation:**

The method is tested on 3 datasets widely used in the literature, IBB (of network traffic in Instanbul), PEMS (similar, but in California), and CiteSeer (scientific citations) with a heat diffusion, and compares it against multiple baselines taken both from the classical Network Theory literature (Erdős–Rényi and Barabási–Albert models) and more modern Neural Network-based methods (with representatives from VAEs, GANs, and Diffusion Models), with TANGEM achieving top performance on IBB1 and Pems04, and strong performance on IBB2 and CiteSeer.

**Requested Changes:**

I believe two analysis would greatly strengthen the paper
1. An analysis on the impact of the transformer architecture on the algorithm's performances, does TANGEM scale better than Digress and similar?
2. While the homophilic assumption is pointed out as a limitation, an initial experiment that helps us quantify how big of a limitation it is would surely improve our understanding of the capabilities of the proposed method, in other words, an experiment where the evolution of the features is not similar between connected nodes, but maybe is dictated by other constraints (number of degrees, whether the node is part of specific graph structures, etc) to shed a light on how much does TANGEM depend on this assumption.

---

> ### Author Response · Authors · 2026-02-26
>
> Thank you very much for the thoughtful and constructive feedback. We are encouraged by your recognition of the paper's scope and appreciate the opportunity to clarify these two critical points.
>
> ## Requested Changes:
>
> **R1:**
> We agree that scalability is an important aspect.  In the revised manuscript, we have included a comparative analysis of TANGEM against DiGress across graph sizes ranging from 50 to 3,200 nodes.
>
> The results show that the quadratic time ($O(T \times n^2)$) and space ($O(n^2)$) complexities of DiGress make it impractical for graphs with more than a few hundred nodes. Conversely, TANGEM operates with $O(M \cdot n)$ time and $O(n)$ space complexity (where $M$ is the number of sampled nodes/tokens). This allows TANGEM to scale to larger graphs while requiring significantly fewer computational resources.
> Details on the complexity analysis have been added to the 'Experiments' section.
>
> **R2:** To quantify the impact of the homophilic assumption, we have added a new experiment (RQ3) evaluating TANGEM across random, homophilic, and heterophilic settings. We measured the alignment between the walker’s trajectory and the underlying graph structure using Normalized Mutual Information (NMI), Adjusted Rand Index (ARI), and Silhouette scores.
>
> The results and details are shared in the revised manuscript 'Experiments - RQ3', and suggest that features misaligned with the graph structure guide the walker to take steps that conflict with the underlying structure (e.g., community), and result in lower clustering performance. In contrast, the homophily case yields the best clusters, while random features experience a slight drop in performance.
>
> This experiment demonstrates that the proposed feature-guided biased random walk is highly effective when the graph homophily is leveraged. Consequently, while the current distance-based metrics (e.g., cosine, Euclidean, DTW) are optimized for homophilic structures, their usage in high-heterophily settings introduces a feature-topology conflict. We have identified the integration of learnable distance metrics rather than distance-based metrics as a key direction for future work to extend the method’s robustness to heterophilic graphs. We have also emphasized that fact in the revised paper.

---

### Review · Reviewer_ahw4 · 2026-03-01

**Summary Of Contributions:**

The paper proposes TANGEM, a graph generator for temporally attributed graphs with fixed topology. The core idea is to compute a pairwise temporal similarity matrix from node time series and use it to bias Node2Vec-style random walks, then train a decoder-only transformer to autoregressively generate node sequences that are converted back into graph structure. The intended contribution is to couple temporal co-activation patterns with structural sampling, rather than modeling topology evolution directly.

Strengths

1. The paper targets a less-studied setting: graph generation for fixed-topology with evolving node attributes, which is at least a plausible and potentially useful.
2. The method is lightweight relative to adjacency-matrix-based diffusion baselines, and the scalability discussion is a practical positive.

Weaknesses

1. The presentation is hard to follow: terminology is inconsistent, the setup is easy to misread, and there are visible writing issues/typos.
2. The problem setup is ambiguous: the paper says it is for fixed-topology graphs, yet it also claims to “generate graphs” from walks, which blurs whether it is generating new topology, reweighting/recovering structure, or primarily exploiting temporal node attributes.
3. The experimental comparison is not fully convincing because several baselines that require multiple training graphs are trained on augmented variants of a single graph, which is not clearly a fair or meaningful comparison to their intended use.
4. The technical novelty appears limited: the method mainly combines known ingredients (biased random walks, similarity-based biasing, autoregressive sequence modeling) and provides little theoretical justification beyond intuition.

**Audience:**

Yes

**Audience Explanation:**

Researchers interested in graph generation, temporal graph learning, or traffic/sensor-network settings with fixed node sets may find the paper interesting.

**Broader Impact Concerns:**

I do not see any major ethical concerns.

**Claims And Evidence:**

No

**Claims Explanation:**

The empirical results do suggest that TANGEM can achieve strong structural MMD scores on the selected benchmarks, and the paper reports improvements over several baselines on those metrics. However, I do not find the evidence fully convincing for several reasons. Refer to the weaknesses above.

**Requested Changes:**

1. Clarify the task definition and outputs. The paper must state unambiguously, early and consistently, what the task is.
2. Substantially improve the writing and proofreading. The manuscript needs careful editing for grammar, notation consistency, and terminology.
3. Rework the baseline comparison to ensure fairness. The current setup trains multi-graph baselines on artificially augmented versions of a single graph.
4. Strengthen the novelty discussion and position the work more carefully. The authors should explicitly distinguish what is new versus what is inherited from prior work.
5. Add stronger analytical or theoretical support. Even a limited theoretical result would help: e.g., analysis of how the proposed bias changes transition behavior, coverage/repetition properties, or what distributional object is being approximated.

---

> ### Author Response · Authors · 2026-03-12
>
> Thank you very much for the thoughtful feedback. We have revised the manuscript to address your concerns.
>
> ### Weaknesses
>
> **W1 & W2 (Clarity and Definitions)**: We’ve updated the definition at the beginning of the **Methodology** section. We’ve clearly defined the task and expected input-outputs. We have also corrected inconsistent terminology and typos throughout the manuscript. (See R1 and R2)
>
> **W3 (Baseline Comparison)**: We thank the reviewer for pointing this out. Comparing TANGEM (primarily designed for (possibly) large-scale spatio-temporal graphs) with models like DiGress (designed for datasets of many small graphs) presents a fundamental challenge. To ensure a fair evaluation, we implemented a controlled augmentation strategy:
>
> - Since GraphRNN, GRAN, and DiGress require a distribution of graphs for robust training, we employed a controlled augmentation strategy by applying low probability edge dropping ( ~3% depending on the network). While doing that, we ensure that the graph’s core structure remains unaffected (e.g., augmentation does not harm the connectivity of network) and preserve high-level patterns such as paths, grids, and communities. This distributional variety is necessary to make these baselines directly comparable to TANGEM.
>
> **W4 (Novelty)**: While TANGEM is built on ingredients such as biased random walks and Transformer architecture, its primary contribution is the explicit integration of a feature-based similarity matrix as a ‘semantic filter’, while the traditional generators are ‘blind’ to non-structural information. While doing that, we provide autoregressive modeling, which allows TANGEM to generate large, semantically consistent graph structures. (See R5)
>
> ### Requested Changes:
>
> **R1**: We've updated the beginning of the **Methodology** section with a clear definition of task.
>
> **R2**: We’ve reviewed the manuscript end-to-end to fixed typos and ensure notation consistency.
>
> **R3**:  We've expanded the explanation of the augmentation strategy that aims to increase fairness.
>
> **R4**: We’ve added a detailed paragraph at the end of the Literature Review section, clarifying the primary contributions and practical utilities of TANGEM. Additionally, we’ve added an ablation study (RQ5) that provides a direct scalability comparison between TANGEM and DiGress, demonstrating TANGEM’s ability to handle large-scale networks. Finally, we’ve added a downstream utility (RQ4), representing an area where we believe TANGEM can be practically applied (data augmentation, network simulation, anomaly detection).
>
> **R5**: We've extended the 'Experiments' section to include RQ3 - Sensitivity to Feature Distribution. This analysis assesses: (1) the noisy edge filtering effect, and (2) behavior in different settings, including homophilic, random, and heterophilic.

---

### Decision · Action_Editor_oPUv · 2026-04-07

**Recommendation:** Accept as is

**Audience:**

Yes

**Audience Explanation:**

Reviewers unanimously agree this is the case.

**Claims And Evidence:**

Yes

**Claims Explanation:**

A majority of reviewers agree this is the case.